# Application of Green Synthesized MMT/Ag Nanocomposite for Removal of Methylene Blue from Aqueous Solution

Nisha Choudhary [1], Virendra Kumar Yadav [1,2], Krishna Kumar Yadav [3], Abdulaziz Ibrahim Almohana [4], Sattam Fahad Almojil [4], Govhindhan Gnanamoorthy [5], Do-Hyeon Kim [6], Saiful Islam [7], Pankaj Kumar [8] and Byong-Hun Jeon [6,*]

[1] Research and Development Centre, YNC ENVIS PVT. LTD., New Delhi 110059, India; nishanaseer03@gmail.com (N.C.); yadava94@gmail.com (V.K.Y.)

[2] Department of Microbiology, School of Sciences, P P Savani University, Surat 394125, India

[3] Faculty of Science and Technology, Madhyanchal Professional University, Ratibad, Bhopal 462044, India; envirokrishna@gmail.com

[4] Department of Civil Engineering, College of Engineering, King Saud University, P.O. Box 800, Riyadh 11421, Saudi Arabia; aalmohanna@ksu.edu.sa (A.I.A.); salmojil@ksu.edu.sa (S.F.A.)

[5] Department of Inorganic Chemistry, Guindy Campus, University of Madras, Chennai 600025, India; gnanadrdo@gmail.com

[6] Department of Earth Resources and Environmental Engineering, Hanyang University, Seoul 04763, Korea; kimdohyeon@hanyang.ac.kr

[7] Civil Engineering Department, College of Engineering, King Khalid University, Abha 61411, Saudi Arabia; sfakrul@kku.edu.sa

[8] Integrated Regional Office, Ministry of Environment, Forest and Climate Change (MoEFCC), Government of India, Saifabad, Hyderabad 500004, India; pankajb434@gmail.com

* Correspondence: bhjeon@hanyang.ac.kr

**Abstract:** Textile industries are the largest consumer of synthetic dyestuff compounds and consequently, they are the prime contributor of colored organic contaminants to the environment. The dye compounds when released in soil or freshwater resources such as rivers, cause a potential hazard to living beings due to their toxic, allergic and carcinogenic nature. Current conventional treatment methods for removal or degradation of such dyestuff materials from water systems are not sufficient, and therefore, there is an immediate need to find efficient and eco-friendly approaches. In this regard, nanotechnology can offer an effective solution to this problem. In the present work, montmorillonite/silver nanocomposite (MMT/Ag nanocomposite) is developed through green synthesis methods using naturally occurring montmorillonite (MMT) clay and silver nanoparticles. The material was characterized by using a particle size analyzer (PSA), UV/Visible spectroscopy, Fourier transform infrared spectroscopy (FTIR), field emission scanning electron microscope (FE-SEM), energy dispersive X-ray (EDX) spectroscopy and a Brunner–Emmett–Teller (BET) surface area analyzer. The adsorption efficiency of the nanocomposite and per cent removal of methylene blue (MB) was investigated by using a batch system.

**Keywords:** montmorillonite; silver; nanocomposite; methylene blue; remediation

## 1. Introduction

Human's fascination with "color" has brought a huge revolution to the industrial sector. The textile industry majorly contributes to water pollution by releasing a huge amount of dyestuff as effluent. This untreated or partially treated industrial effluent containing synthetic dye compounds is eventually discharged into water systems, where it causes damage to the aquatic ecosystem [1,2]. Some metal complex-based synthetic dyes are causing severe mutagenic and other health issues even at ppm concentrations [3]. Methylene blue (MB) is one of the most widely used cationic dyes in biological staining and in textile industries because of its faster adsorption on cotton fabric, water solubility

and economic benefits. However, MB has been reported to cause diarrhea, eye irritation, vomiting, breathing difficulty, nausea and chronic toxicity in humans, mainly in relation to the central nervous system [4,5]. Therefore, the removal of MB dye molecules from aqueous systems has become a crucial requirement. MB ($C_{16}H_{18} N_3SCl$) is a heterocyclic aromatic compound with 319.85 g/mol of molecular weight. MB is a cationic dye molecule that has a positive charge and its chemical structure is shown in Figure 1.

**Figure 1.** Chemical structure: MB chloride salt.

Various physicochemical and biological treatment methods are available for the removal of MB dye molecules from the water system [6,7]. Further, various conventional methods such as coagulation, flocculation [8], oxidation/ozonation [9], membrane-based separation [10] have been used to separate dye molecules from wastewater [11]. However, there are some limitations with these methods such as high cost, toxic intermediate production, and lower efficiency. Therefore, adsorption techniques are being used by industries because of their low cost and easy handling [12–14].

Numerous adsorbents (natural and engineered) have been studied by various researchers for the removal of dye molecules such as activated carbon [15,16], zeolite [17], tourmaline [18], black phosphorous nanosheet [19], waste materials [20], dead biomass [21], natural clay and clay minerals [22,23] and agricultural waste materials [24]. Recently, nanotechnology-driven materials such as clay/metal nanocomposites are being explored for their potential as efficient adsorbents, where metal nanoparticles act as fillers and clay provides support for their synthesis. Clay is an abundantly available, natural and low-cost material with excellent cation exchange efficiency due to its high surface area, mechanical stability, swelling properties and high cation exchange capacity (CEC) [25]. Montmorillonite (MMT) is one such clay mineral that has been studied for its ability to adsorb different types of dye compounds [26]. Silver nanoparticles are used for catalytic degradation of textile dyes [27].

Some previous studies have reported the synthesis of MMT clay and silver nanocomposite by the chemical reduction method [28], γ-irradiation technique [29], and green methods using plant extracts of medicinal plants [30–32] and fruit waste [33]. These studies have focused on the antimicrobial property of the nanocomposite. In the present study, we have developed montmorillonite clay/silver (MMT/Ag) nanocomposite as an adsorbent material for the removal of MB dye from its liquid solution, keeping in mind that clay has a high cationic exchange capacity, high absorption capacity and silver nanoparticles are also known to have good catalytic efficiency [34]. Besides, this is the first time we are reporting the use of a weed plant, *"sida acuta"* extract for the synthesis of an MMT/Ag nanocomposite where MMT clay is used as a matrix and silver nanoparticles as nanofillers. *Sida acuta* Burm. f., also known as broom weed, is a perennial undershrub of the Malvaceae family. Although, the weed plant has its native roots in Central America, it is also widely distributed in pan-tropical regions of the world including India. *S. acuta* is used for the treatment of diseases related to the liver, blood, and the urinary and nervous systems [35]. This plant contains bioactive compounds, thus it has numerous pharmacological activities such as anti-ulcer, analgesic, antipyretic, antiviral, anti-plasmodial, anticancer, antimicrobial and anti-inflammatory properties [36]. The use of the extract from the weed leaves is a rapid, economic and environmentally-friendly method for nanomaterial synthesis.

The phytochemicals and bioactive compounds in the aqueous leaf extract of *S. acuta* act as a reducing agent for silver salt and result in the formation of silver nanoparticles

in the lamellar space of MMT clay [32]. The clay acts as a matrix in the composite, which provides support and also prevents the agglomeration of silver nanoparticles during the synthesis process. The developed nanocomposite and raw MMT clay were characterized for their morphology and structural information. The PSA provided information related to the particle size of nanocomposite material whereas FTIR analysis was carried out to observe the functional groups in the sample. FE-SEM analysis was done to study the surface features, EDX for elemental analysis, and BET surface area analysis was conducted to evaluate the specific surface area. Finally, the adsorption potential of the developed nanocomposite was studied by conducting batch adsorption experiments for the removal of MB dye from an aqueous solution.

## 2. Materials and Methods

### 2.1. Reagents and Materials

Silver nitrate ($AgNO_3$) with 99.00% purity and montmorillonite (MMT) clay used for the study were purchased from Merck. *Sida acuta* leaves were collected from the Gandhinagar region, Gujarat, India. The solutions were prepared using distilled water (DW).

### 2.2. Synthesis of MMT/Ag Nanocomposite

Freshly collected *Sida acuta* leaves were washed by thorough rinsing with DW to ensure complete removal of unwanted particles and were allowed to dry in the shade. Then, 10 gm of dry *S. acuta* leaves were boiled in 100 mL of DW at 100 °C for 5 min as shown in Figure 2.

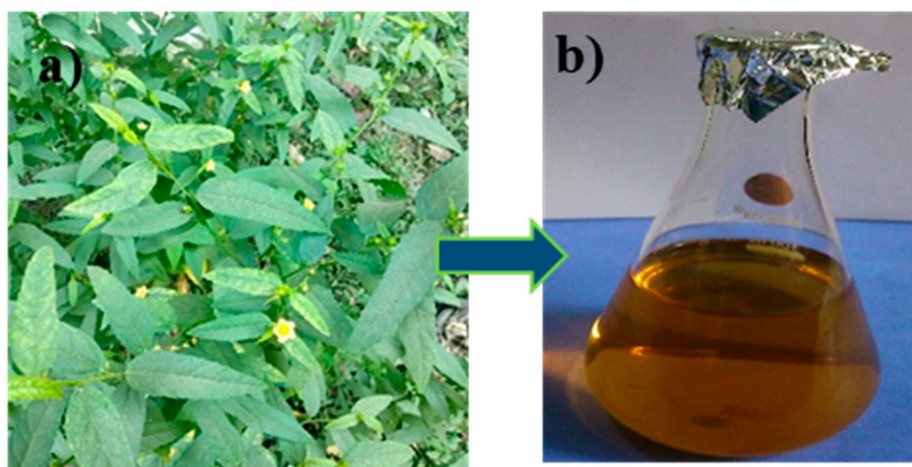

**Figure 2.** *Sida acuta*: (**a**) Plant and (**b**) Leaf extract.

The leaf extract was allowed to cool to room temperature and then filtered to remove pigments and other insoluble fractions. This fresh extract was used for nanocomposite synthesis. For nanocomposite preparation, 10 gm of MMT clay was dried in a hot air oven at 50 °C for 30 min. Figure 3 presents the synthesis procedure for the development of MMT/Ag nanocomposite.

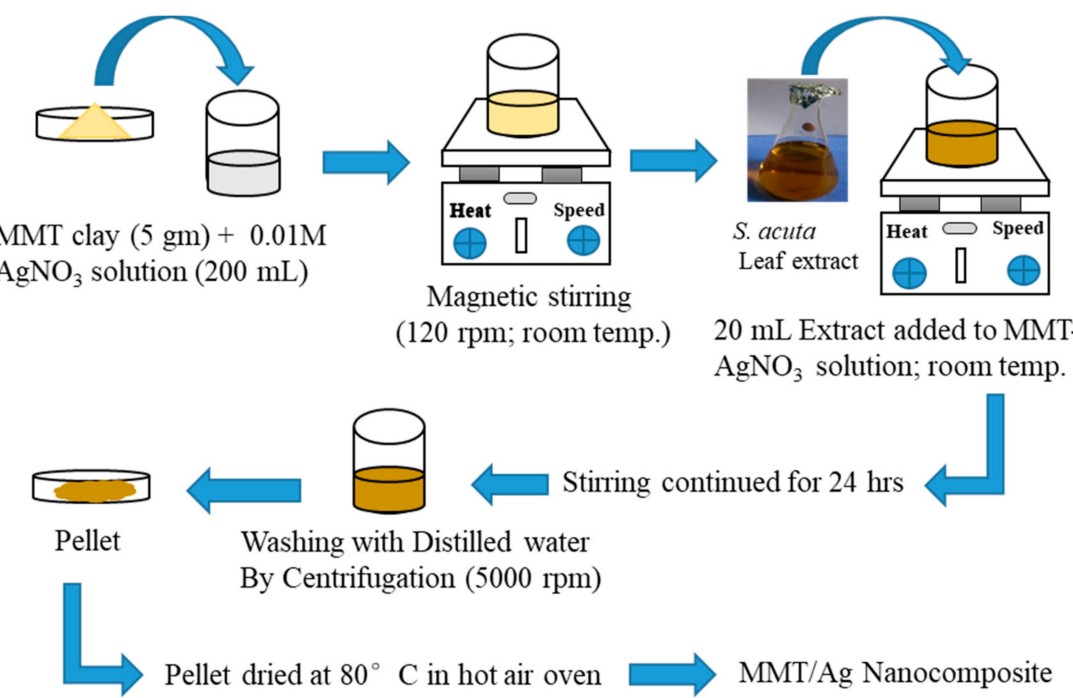

**Figure 3.** Synthesis process of MMT/Ag nanocomposite.

### 2.3. Batch Adsorption Experiments

Batch experiments were used to study the effects of experimental parameters such as different concentrations of MB (25, 50 100 and 200 ppm), contact time, adsorbent dosage (25, 50, 100 mg in 50 mL of MB solution) and the initial dye concentration on the absorption of the dye by the nanocomposite. For these experiments, 250 mL conical flasks with 50 mL of dye solution were used. To determine the adsorption capacity of MMT/Ag nanocomposite, 50 mL of MB solution with varying concentrations (25 ppm, 50 ppm, 100 ppm, and 200 ppm) was prepared. An assessment of different adsorbent doses was conducted with 50 mL of 50 ppm dye solution and 50 mg of MMT/Ag nanocomposite was added to the MB solutions. The mixtures were agitated mechanically in a rotary shaker at 150 rpm for 30 min. After a 5 min interval, aliquots of 3 mL reaction mixture were centrifuged at 5000 rpm for 5 min. Further, the supernatant solution was examined for the remaining MB concentration by using a UV spectrophotometer ($\lambda_{max}$ = 665 nm). The per cent removal of dye and the adsorption capacity of adsorbent at any time ($q_t$) and at equilibrium ($q_e$) were analyzed using the following equations.

$$\% \text{ Removal} = \frac{C_0 - C_t}{C_0} \times 100 \tag{1}$$

$$q_t \text{ (mg|g)} = \frac{(C_0 - C_t)V}{M} \tag{2}$$

$$q_e \text{ (mg|g)} = \frac{(C_0 - C_e)V}{M} \tag{3}$$

where:

$C_0$ = initial MB concentration in mg/L;
$C_t$ = MB concentration at time $t$, in mg/L;
$Ce$ = MB concentration at equilibrium, in mg/L
$V$ = volume of MB solution in liters;
$M$ = mass of adsorbent material in grams (g).

### 2.4. Characterization Techniques

Both MMT clay and MMT/Ag nanocomposite were analyzed by FTIR for the identification of functional groups, for which 2 mg powder samples were mixed with 98 mg of KBr to obtain a solid pellet. The analysis was done by using the Perkin Elmer "Spectrum 6500" (USA). The measurement was done in the band region of 400–4000 cm$^{-1}$, with a resolution of 1 nm. The particle size distribution of both MMT and MMT/Ag nanocomposite was analyzed by a particle size analyzer, for which liquid formulations was prepared by dispersing 1–2 mg powders in double-distilled water (ddw) and sonicating for 10 min in an ultrasonicator (Sonar, 40 Khz, Gujarat, India). The sonicated dispersed samples were divided into two portions, one portion was used for the PSA measurement, which was done by the Malvern Zetasizer, Z-90 (Malvern Instruments Ltd, Malvern, UK) at room temperature (RT), while the second portion of the dispersed sample was used for the UV-Vis measurement in the range of 200–700 nm, using a 2060+ Analytical double-beam spectrophotometer (Analytik Jena Japan Co., Ltd., Yokohama Kanagawa, Japan). For FE-SEM analysis of the powder samples, a pinch of the samples was spread on a carbon tape, which was fixed on the aluminum stub. Both the samples were subjected to gold sputtering. FESEM analysis revealed the morphological properties of the samples, and the measurement was done by using the Nova NanoSEM 450 (FEI Company. Hillsboro, OR, USA), while the elemental analysis of the samples was done by using the attached Bruker-made EDX analyzer. A BET analyzer was used for the measurement of the surface area and the pore size of both the powder samples NOVA 1200e (Quanta Chrome Instruments, Vernon Hills, IL, USA).

## 3. Results and Discussions

### 3.1. PSA for Particle Size Distribution

The particle size distribution of both MMT and MMT/Ag nanocomposite was analyzed by a particle size analyzer, for which liquid formulations were prepared by dispersing 1–2 mg powders into the DW and sonicating for 10 min in an ultrasonicator (Sonar, 40 KHz). The PSA graph of both the particles exhibits two types of species, i.e., it is a bimodal graph as shown in Figure 4. The majority of the MMT particles are below 1000 nm, while some of the particles fall in the size range of 3000–4000 nm. The average size of the MMT particle is 341.3 nm (dm) and the zeta potential value is −37.13 mv.

In the case of MMT/Ag nanocomposite, the majority of the particles also vary in size from 100 nm to 1000 nm, while the size of the particles ranges from 4500 to 6000 nm in the total population. The average size of the MMT/Ag nanocomposite particles is 366.2 nm and the zeta potential is −16.49 mv.

So, based on the PSA, it is clear that the size of the MMT/Ag nanocomposite increased slightly, i.e., 15–20 nm, as the silver nanoparticles were deposited on the surface of the clay, which is also evidenced by the microscopic analysis and XRD. The zeta potential was decreased in the case of the nanocomposite, which may indicate the instability of the samples in the solution as compared to clay, as the deposition or loose attachment of the silver nanoparticles on the surface of the clay might have made them unstable in the distilled water. Gashti et al. also reported that the presence of silver nanoparticles on the surface of clay (kaolin) decreased the thermal stability of the material due to proton delocalization of hydroxyl groups in the nanocomposite [37].

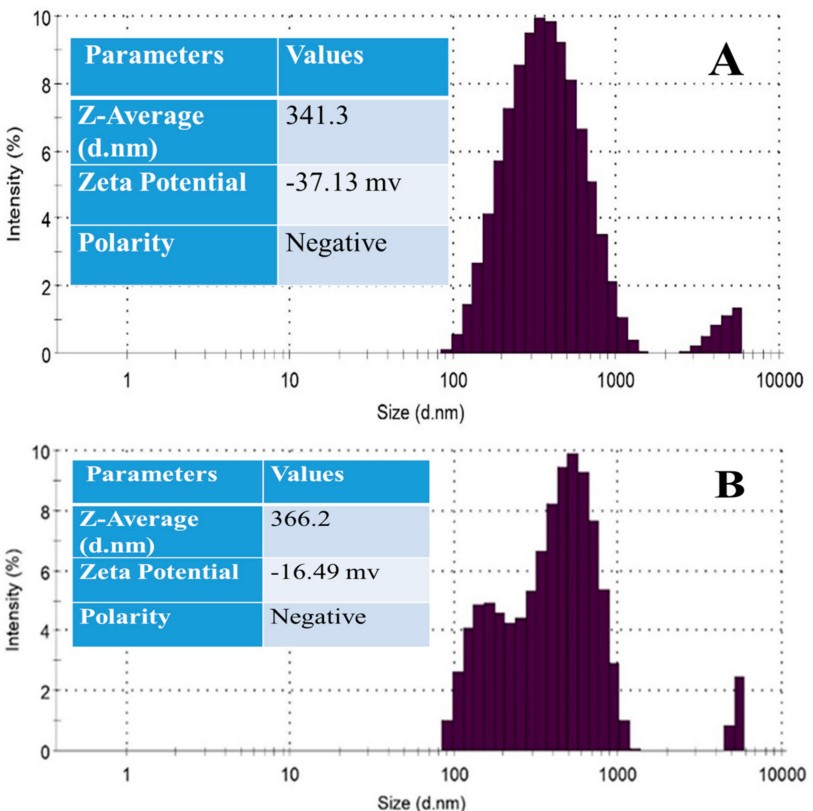

**Figure 4.** PSA graph of (**A**) MMT clay and (**B**) MMT/Ag nanocomposite.

### 3.2. UV-Visible Spectroscopy Analysis

Light absorption by the molecules in the UV or near UV region of the electromagnetic spectrum results in electronic transitions from the ground state to a state of higher energy. This happens because of a high degree of conjugation of the molecules. The UV-Vis spectra for MMT clay and MMT/Ag nanocomposite are shown in Figure 5 with peaks ranging from 200–400 nm.

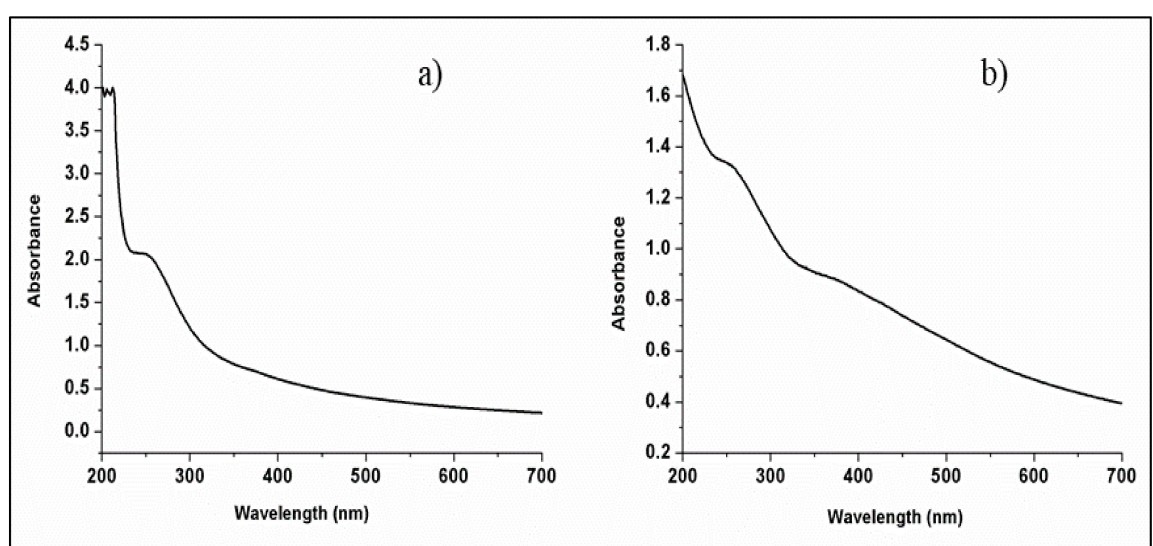

**Figure 5.** UV-Vis spectra: (**a**) MMT clay and (**b**) MMT/Ag nanocomposite.

In the MMT clay sample, the characteristic bands are present in the ultraviolet region (200–400 nm), i.e., at 210 nm, 214 nm and 250 nm. Bands at 210 nm and 214 nm point to the

existence of exchange sites and iron valency in the lattice. These absorption bands display the charge translocation of oxo-to-iron (III) [38]. Also, the EDX analysis of the clay confirms the presence of iron in its elemental composition. The absorption peak at 250 nm is due to $\pi$-to-$\pi^*$ electronic transition of the functional group Si=O. The vibrational frequencies of group Si=O were also detected in the FTIR spectra of MMT clay. A similar study regarding UV spectroscopic analysis of MMT clay has been reported by Wanyika et al. [39]. The absorption spectrum of MMT/Ag nanocomposite shows a low-intensity band at 365 nm, which confirms the presence of silver nanoparticles in the matrix of MMT clay.

### 3.3. FTIR Analysis

The FT-IR analysis showed that spectra (Figure 6) for MMT and MMT/Ag nanocomposite were quite similar with a slight shift in the band's position and intensity. In both samples, Si-O and Al-OH were the main functional groups detected in the range of 1000 $cm^{-1}$ and 500 $cm^{-1}$.

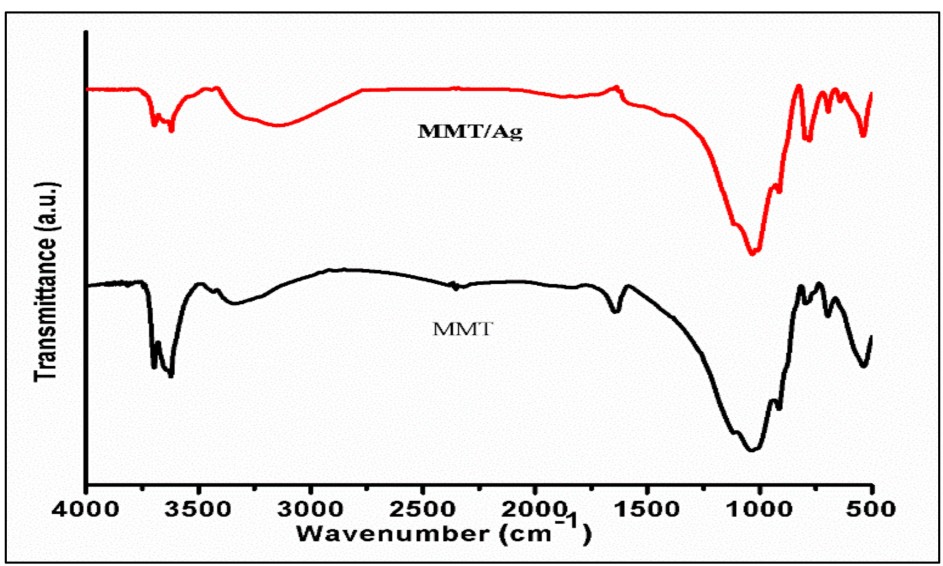

**Figure 6.** FTIR spectra of MMT and MMT/Ag nanocomposite.

The absorption peak at 528 $cm^{-1}$ and 790 $cm^{-1}$ is because of the bending vibration of the Al-O-Si bond and the $Fe^{3+}$-OH-Mg stretch, respectively. Bands at 1130 $cm^{-1}$, and 1048 $cm^{-1}$ were attributed to Si-O stretching vibrations. IR peaks at 3626 $cm^{-1}$ and 918 $cm^{-1}$ were ascribed to the Al-Al-OH group of MMT clay [40], while bands at 3710 $cm^{-1}$, 3448 $cm^{-1}$ and 1675 $cm^{-1}$ were assigned to the O-H functional group, which is due to the existence of water molecules between the layers of MMT. Similar FTIR spectral results for MMT were found by Wanyika et al. The IR spectrum for the MMT/Ag nanocomposite was similar to that of MMT with peaks or bands at the same location but with altered intensities. For instance, the IR band at 918 $cm^{-1}$ moved slightly to 914 $cm^{-1}$ with an enhanced intensity, which indicates an increase in the mass of the molecules due to silver nanoparticles-MMT clay interaction. The MMT/Ag nanocomposite spectra showed a new absorption band near 3150 $cm^{-1}$, which is an indication of the interaction between silver nanoparticles and the O-H functional group in the MMT layers [29,41].

### 3.4. FESEM, TEM and EDX Analysis

The analysis of surface structures and elemental composition of MMT clay and MMT/Ag nanocomposite was done using FESEM and EDX (Figure 7). The results suggest that the nanocomposite has a flaky and layered structure that contains silicon, aluminum and silver as the principal constituents.

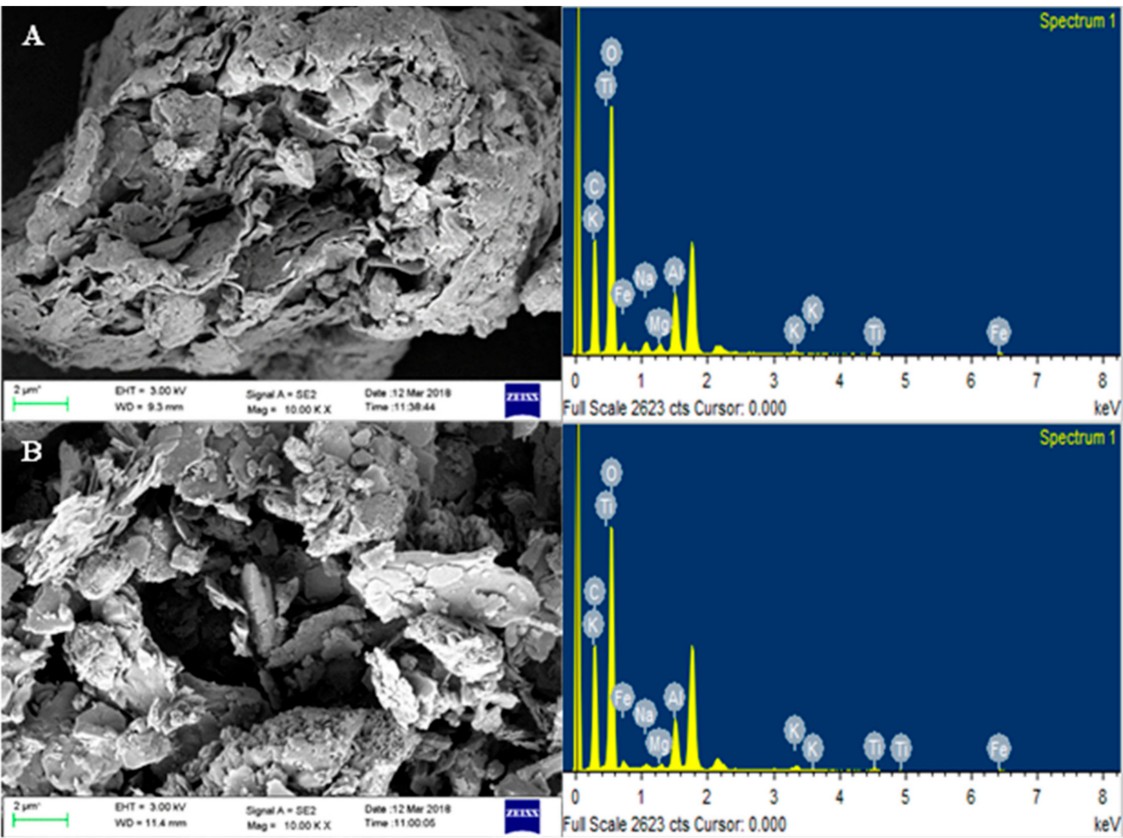

**Figure 7.** FESEM images and EDX spectrum of (**A**) MMT clay and (**B**) MMT/Ag nanocomposite.

The FESEM images of MMT/Ag nanocomposite shows the presence of silver nanoparticles on large flakes of MMT clay, confirming the interaction between the two. Also, the interaction between MMT and silver nanoparticles in the nanocomposite has already been proven by the FTIR results. The elemental composition of the nanocomposite obtained by EDX showed that it contains Si (31.55%), Al (6.02%), Fe (8.86%), and Ag (1.38%). Hence, microscopic analysis outcomes suggest that the nanocomposite is a heterogeneous composite containing flakes of MMT and silver nanoparticles.

*3.5. BET Surface Area Analysis*

The specific surface area and porosity of MMT clay and MMT/Ag nanocomposite were analyzed by using the BET analyzer model no. NOVA-1200e (Quanta Chrome Instruments). Surface area analysis by BET involves the physisorption of gas (nitrogen) onto the sample surface with weak van der Waal force. The Barrett–Joyner–Halenda (BJH) method was used for the evaluation of pore volume and pore diameter. The results are shown in Table 1. The surface area of MMT/Ag nanocomposite was found to be higher than raw MMT clay. The nanocomposite specific surface area was found to be 82.663 $m^2$/g with a pore volume and pore diameter of 0.127 cc/g and 4.136 nm, respectively. Smaller pore size and higher specific surface area are characteristic features of an efficient and highly capable adsorbent [42].

**Table 1.** Surface area of MMT and MMT/Ag nanocomposite.

| Material | Surface Area ($m^2$/g) | Pore Volume (cc/g) | Average Pore Diameter (nm) |
|---|---|---|---|
| Montmorillonite Clay (Raw) | 77.234 | 0.116 | 3.782 |
| MMT/Ag nanocomposite | 82.663 | 0.127 | 4.136 |

### 3.6. Adsorption Results of MB Using MMT/Ag Nanocomposite

The adsorption system was evaluated by examining the results of various experimental parameters such as the effect of contact time, initial MB concentration and nanoadsorbent dose.

### 3.6.1. Effect of Contact Time

The removal of MB from aqueous solution occurred either via a physisorption or chemosorption process. During 30 min of contact time, the amount of MB adsorbed onto the nanocomposite was found to increase with time up to 20 min, and then it attained equilibrium. A quick decrease in the MB concentration during the initial 5 min shows strong contact between the cationic dye MB and the negative charge-carrying nanocomposite material. After 5 min, the MB adsorption increased gradually up to 20 min. The per cent removal of four different concentrations of MB using a fixed amount of raw clay and adsorbent, up to 30 min of contact time is shown in Figure 8. The figure shows that MB removal efficiency (for all four concentrations) increased with contact time whereas the per cent removal of MB decreased with the increase in MB concentration. After 30 min of contact time, the MMT/Ag nanocomposite showed MB removal of 99.90% for 25 ppm, 96.50% for 50 ppm, 89% for 100 ppm and 81.14% for 200 ppm. Similar results were reported by [43] for MB dye adsorption by montmorillonite clay.

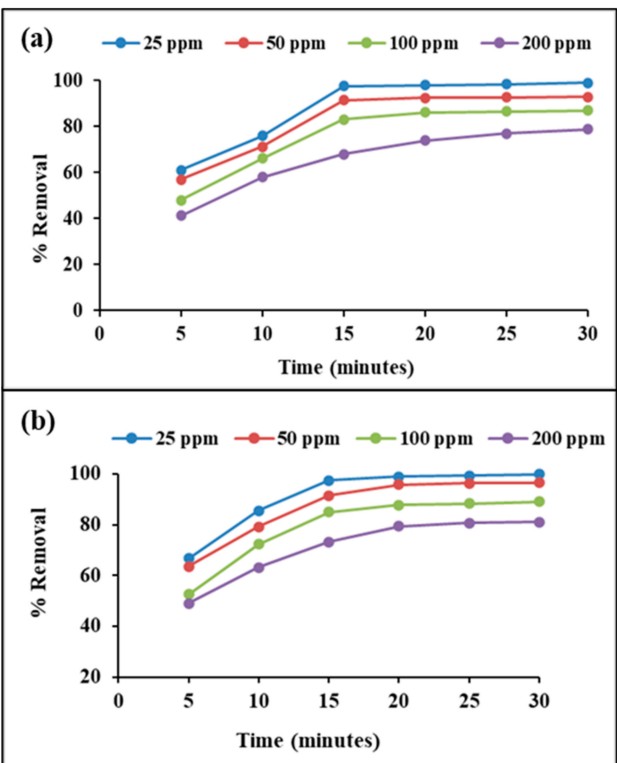

**Figure 8.** Removal percentage of MB using (**a**) MMT clay and (**b**) MMT/Ag nanocomposite.

### 3.6.2. Effect of Initial MB Concentration

Figure 9 shows the effect of increasing the initial MB concentration on the adsorption efficiency of the raw MMT clay and nanocomposite. The removal efficiency of both materials decreased with increasing MB concentration. The reason could be the unavailability of sufficient active sites on the nanocomposite surface for binding with MB ions. Similar observations were reported by [44] for MB adsorption on red clay.

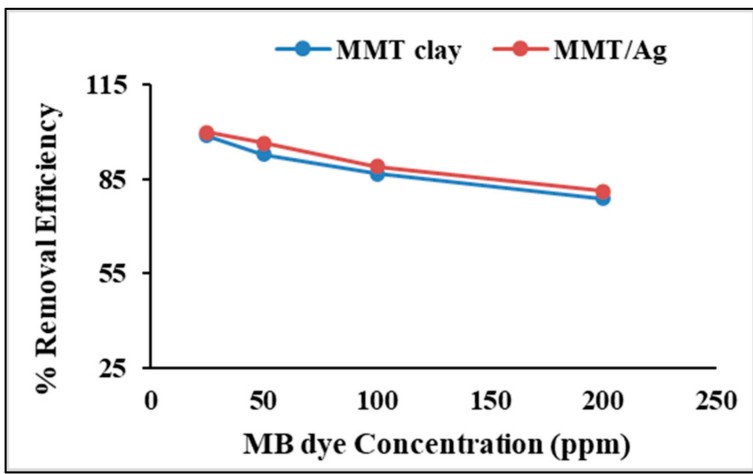

**Figure 9.** Effect of initial MB concentration on adsorption by raw MMT clay and MMT/Ag nanocomposite.

### 3.6.3. Effect of Adsorbent Dose

To identify the optimum adsorbent dose, the effects of adsorbent dosage was studied with regard to the amount of MB concentration. In general, increasing the adsorbent dose increases the removal efficiency. Figure 10 shows an increase in the removal efficiency of adsorbents, which is due to the easy and excessive availability of adsorption sites at the nanocomposite surfaces. Increases in the removal efficiency slowed down after 50 mg, which suggested that 50 mg is the optimum nanocomposite dose for adsorption of 100 ppm MB dye from an aqueous solution. The obtained outcome is in good agreement with the results reported by [45].

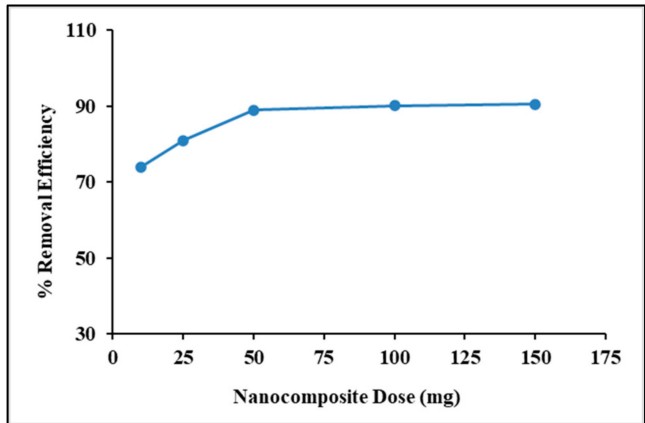

**Figure 10.** Effect of adsorbent doses on MB removal efficiency.

### 3.6.4. Effect of Contact Time on Adsorption Capacity of Nanocomposite

The variations detected in MB adsorption on MMT clay (raw) and MMT/Ag nanocomposite as a function of contact time are shown in Figure 11a–d. The figure shows that the maximum MB adsorption occurred during the initial 15 min. Afterwards, it increased gradually and almost reached equilibrium within 30 min of the initial contact. The faster adsorption rate in the initial time period is because vacant and available sites on the negatively-charged adsorbent surface result in faster adsorption of negative charge-carrying MB dye from the aqueous solution at neutral pH [46,47]. Further, the decreased adsorption rate is due to a lack of available active sites resulting from MB monolayer formation on the adsorbent surface. Similar results were also reported by Vutskits et al. [5] for MB adsorption by activated carbon. The adsorption capacity of raw MMT clay ranged from 13 mg/g to 151.68 mg/g and MMT/Ag nanocomposite ranged from 14.32 to 156.22 mg/g against adsorption of MB dye [48].

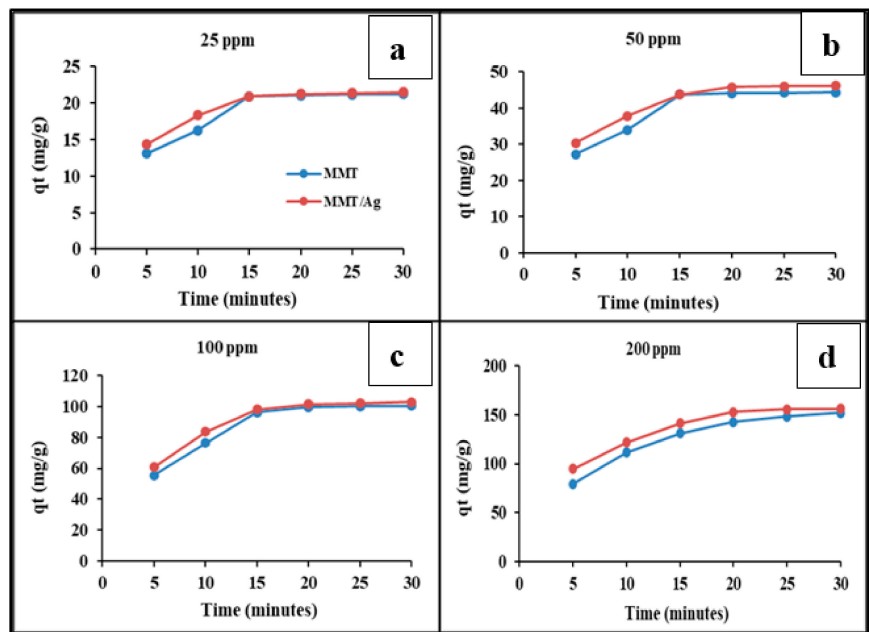

**Figure 11.** Effect of the contact time on adsorption capacity of raw MMT clay and MMT/Ag nanocomposite (**a**) 25 ppm, (**b**) 50 ppm (**c**) 100 ppm and (**d**) 200 ppm of dyes.

## 4. Conclusions

The outcomes of the present research study show that MMT/Ag nanocomposite was successfully synthesized using a weed plant extract and effectively used as an adsorbent for the removal of cationic dyes from an aqueous medium. The FTIR results showed the interaction between the clay and silver nanoparticles and the FESEM study revealed the layered, sheet-like morphology of MMT/Ag nanocomposite. The MB adsorption study showed that both raw MMT and MMT/Ag nanocomposite have potential for the removal of MB dye and the difference in their adsorption efficiency was found to be very small. This might be due to the low silver content in the developed nanocomposite material. Further research should be conducted with different clay: Ag ratios in order to evaluate the adsorption capacity of clay/Ag nanocomposite against the adsorption of MB dye. Silver-based clay nanocomposites are well-known antimicrobial agents. More research on this material could provide a solution for the removal of microbial and organic pollutants from wastewater.

**Author Contributions:** Conceptualization, A.I.A., V.K.Y., S.F.A., G.G. and A.I.A.; Data curation, N.C., A.I.A., S.I. and S.F.A.; Methodology, N.C., G.G., D.-H.K. and P.K.; Validation, K.K.Y., S.I., N.C., B.-H.J. and D.-H.K.; Formal analysis, V.K.Y. and A.I.A.; Resources, B.-H.J. and K.K.Y.; Writing—original draft preparation, N.C.; Writing—review and editing, V.K.Y., K.K.Y., N.C., S.F.A., A.I.A. and S.I.; Supervision, V.K.Y., K.KY. and B.-H.J.; Project administration, A.I.A. and B.-H.J.; Funding acquisition, B.-H.J., S.F.A. and A.I.A.; Investigation, V.K.Y., B.-H.J. and P.K., S.I.; Software, G.G., K.K.Y., P.K. and D.-H.K.; Visualization, P.K., V.K.Y. and D.-H.K. All authors have read and agreed to the published version of the manuscript.

**Funding:** The authors would like to acknowledge the support of the Researchers Supporting Project number (RSP-2021/303), King Saud University, Riyadh, Saudi Arabia. This study was supported by grants from the Korea Institute of Energy Technology Evaluation and Planning (KETEP) funded by the Ministry of Trade, Industry & Energy (MOTIE) of the South Korean Government (No. 20206410100040).

**Conflicts of Interest:** There is no conflict of interest regarding the publication of this paper.

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
