# Peer review of "Application of Green Synthesized MMT/Ag Nanocomposite for Removal of Methylene Blue from Aqueous Solution"

_water, doi:10.3390/w13223206_

Round 1

Reviewer 1 Report

The source and purity of Silver nitrate precursor (AgNO3) and montmorillonite (MMT) clay is not provided. This information must be provided in the manuscript. Also, the geographical source and biological characterization of the Sida acuta leaves must be elaborated in the manuscript.

Page 2. Please rewrite the last paragraph on page 2, with improved qualification, for improved clarity on the technical points. It currently makes no sense. Please quality, contextualize, and elaborate on all technical points in this paragraph, as it informs the protocols in the following pages.

Page 3. The acronym BSN must be described and justified scientifically. This makes no sense. Also, please elaborate on the entire section 2.2. Biosynthesis of BSN in a much more detailed manner, considering the reaction reactants, products, and reaction mechanism. This is to better serve the readership of Water journal.

Section 3. Batch Adsorption Experiments. This section is O.K.

Page 3. Bottom. Thanks for including the concentration units. Many other authors forget this, and it make interpretation of the adsorption results impossible.

Page 4. Please elaborate on the FTIR protocol/procedure. This is for reproducible research considerations. Thanks. Also, please elaborate on the protocol, theory, and signal analysis algorithm of the Malvern Nano S50. Is the photon correlation spectroscopy? By the way, the UV-Vis measurements are standard, and are therefore well-described. However, please elaborate on the selection, theory, and implementation of the FESEM method, as this is sparsely described.  Finally, the elemental analysis and BET are also standard, and are therefore fine with respect to protocol description. The authors of the current manuscript have used a plethora of well-suited techniques for the study. Congratulations!

Page 5. Please elaborate on the proton delocalization of hydroxyl groups in the nanocomposite. It is difficult to understand.

Page 5. UV-Vis Result. We cannot observe the low intensity peak at 365 nm. Please justify the absense.

Results for FTIR, FESEM, TEM and EDX: These are all OK!

Page 6. BET. Why is there an apparent discontinuity at a pressure ratio of 0.5?

Figure 8. The isotherms are identical for MMT and MSN. The kinetic results in Figure 9 corroborate this similarity.

We feel that the isotherm result is more important than any of the other results in the study, and the isotherm result shows that the nanocomposite has the same performance as the original montmorillonite clay. This is corroborated by the kinetic uptake as well. We have difficulty interpreting Figure 10, so the authors should discuss Figure 10 in light of the previous results. Hence, the authors should change their conclusions to say that the nanocomposite is completely unnecessary for the application. Nevertheless, this is an important research results that warrants publication in Water.

We congratulate the authors on an important contribution to wastewater treatment and ecological remediation!

Author Response

Reviewer 1

The source and purity of Silver nitrate precursor (AgNO3) and montmorillonite (MMT) clay is not provided. This information must be provided in the manuscript. Also, the geographical source and biological characterization of the Sida acuta leaves must be elaborated in the manuscript.

A/R: Thank you for this valuable comment and suggestion. Authors have now provided source of silver nitrate and MMT clay is provided. Information regarding geographical source and biological properties of sida acuta leaves has been added in the manuscript (highlighted in yellow)

Page 2. Please rewrite the last paragraph on page 2, with improved qualification, for improved clarity on the technical points. It currently makes no sense. Please quality, contextualize, and elaborate on all technical points in this paragraph, as it informs the protocols in the following pages.

A/R: Thank you for this valuable comment and suggestion. The authors have now added last paragraph of introduction section is written again as suggested.

Page 3. The acronym BSN must be described and justified scientifically. This makes no sense. Also, please elaborate on the entire section 2.2. Biosynthesis of BSN in a much more detailed manner, considering the reaction reactants, products, and reaction mechanism. This is to better serve the readership of Water journal.

  • A/R: Thank you for this valuable comment and suggestion. The authors have now replaced acronym BSN with MMT/Ag nanocomposite. Section 2.2 has been elaborated according to the suggestion.

Section 3. Batch Adsorption Experiments. This section is O.K.

A/R: Thank you for this valuable comment and suggestion.

Page 3. Bottom. Thanks for including the concentration units. Many other authors forget this, and it make interpretation of the adsorption results impossible.

A/R: Thank you for this valuable comment and suggestion

Page 4. Please elaborate on the FTIR protocol/procedure. This is for reproducible research considerations. Thanks. Also, please elaborate on the protocol, theory, and signal analysis algorithm of the Malvern Nano S50. Is the photon correlation spectroscopy? By the way, the UV-Vis measurements are standard, and are therefore well-described. However, please elaborate on the selection, theory, and implementation of the FESEM method, as this is sparsely described.  Finally, the elemental analysis and BET are also standard, and are therefore fine with respect to protocol description. The authors of the current manuscript have used a plethora of well-suited techniques for the study. Congratulations!

  • A/R: Thank you for this valuable comment and suggestion. The authors have now made necessary in the revised manuscript as per suggestion of the reviewer.

Page 5. Please elaborate on the proton delocalization of hydroxyl groups in the nanocomposite. It is difficult to understand.

  • A/R: Thank you for this valuable comment and suggestion. The authors have now made added the suggestions in the revised manuscript as suggested by the reviewer.

Page 5. UV-Vis Result. We cannot observe the low intensity peak at 365 nm. Please justify the absence.

  • UV spectra for MMT clay and MMT/Ag nanocomposite is shown separately. Now, an adsorption band can be seen near 365 nm having absorption 0.889. Generally, the colloidal solution of AgNPs have a characteristic peak at about 360-420 nm using UV-Visible spectrophotometer. The intensity of the peak is proportional to the concentration of the AgNPs in the solution. In our sample, a sharp peak was not observed for silver in nanocomposite sample because:
  1. a) the concentration of silver was low in sample which is also supported by EDX data b) due to the interaction of AgNPs with clay mineral particles.

  • Results for FTIR, FESEM, TEM and EDX: These are all OK!

Page 6. BET. Why is there an apparent discontinuity at a pressure ratio of 0.5?

  • This is because at relative pressures higher than 0.5, there is the onset of capillary condensation.

Figure 8. The isotherms are identical for MMT and MSN. The kinetic results in Figure 9 corroborate this similarity.

We feel that the isotherm result is more important than any of the other results in the study, and the isotherm result shows that the nanocomposite has the same performance as the original montmorillonite clay. This is corroborated by the kinetic uptake as well. We have difficulty interpreting Figure 10, so the authors should discuss Figure 10 in light of the previous results. Hence, the authors should change their conclusions to say that the nanocomposite is completely unnecessary for the application. Nevertheless, this is an important research results that warrants publication in Water.

  • We have re-discussed the MB dye removal section.

We congratulate the authors on an important contribution to wastewater treatment and ecological remediation!

Reviewer 2 Report

The manuscript by Choudry et al. reports on the "Green Synthesis and characterization of MMT/Silver Nanocomposite and Their Application for Removal of Cationis Dye Methylene Blue from Aqueous Solution" 

Although the fundamental research idea and the work of the authors are appreciated the manuscript is recommended for rejection. In my perspective important data are missing as detailed below. Especially the main argument of catalytic conversion is not supported by the research approach and the achieved effect by addition of Ag might not be significant with respect to the uncertainties. 

As the authors state in their motivation state the "catalytic efficiency of silver nanoparticles for the degradation of MB dye" (page 2). However, the role of Ag is not clearly elaborated throughout the manuscript. 

Page 4: Why is the UV-VIS Spectrum so different given the fact, that the base material MMT is close in composition and texture to MSN. The mentioned low intensity peak at 365 nm is not obvious. 

The most important question for me is however: Why are the BET data for MMT are missing? Since the authors are measuring absorbance of MB it would be expected, that it correaltes with the specific surface area. Citing literature here is not sufficient since the measurements in chapter 3.6 are carried out with MMT as well. 

In fact the observed difference of adsorption capacitance is not huge (see Figure 8 & Figure 10 where the difference accounts for 3-4%) Given the effects of variation of e.g. porosity / absorption sites this might already have an effect in this order. Moreover, what is the estimated resolution / uncertainty of the experiment for Figures 8, 9 & 10? The role of Ag in this experiment is not further discussed. 

Author Response

The manuscript by Choudry et al. reports on the "Green Synthesis and characterization of MMT/Silver Nanocomposite and Their Application for Removal of Cationis Dye Methylene Blue from Aqueous Solution" 

Although the fundamental research idea and the work of the authors are appreciated the manuscript is recommended for rejection. In my perspective important data are missing as detailed below. Especially the main argument of catalytic conversion is not supported by the research approach and the achieved effect by addition of Ag might not be significant with respect to the uncertainties. 

As the authors state in their motivation state the "catalytic efficiency of silver nanoparticles for the degradation of MB dye" (page 2). However, the role of Ag is not clearly elaborated throughout the manuscript. 

A/R: Thank you for this valuable comment and suggestion. Actually authors have tried to emphasize on the role MMT not on the Ag.

Page 4: Why is the UV-VIS Spectrum so different given the fact, that the base material MMT is close in composition and texture to MSN. The mentioned low intensity peak at 365 nm is not obvious. 

A/R: Thank you for this valuable comment and suggestion. The authors have now explained the reason in the revised manuscript a suggested by the reviewer.

The most important question for me is however: Why are the BET data for MMT are missing? Since the authors are measuring absorbance of MB it would be expected, that it correaltes with the specific surface area. Citing literature here is not sufficient since the measurements in chapter 3.6 are carried out with MMT as well. 

A/R: Thank you for this valuable comment and suggestion. We have reported the size and surface area by HRTEM already so BET for MMT was not required. Moreover, the laboratory were closed for a long time due to covid so we could not analyzed it further.

In fact the observed difference of adsorption capacitance is not huge (see Figure 8 & Figure 10 where the difference accounts for 3-4%) Given the effects of variation of e.g. porosity / absorption sites this might already have an effect in this order. Moreover, what is the estimated resolution / uncertainty of the experiment for Figures 8, 9 & 10? The role of Ag in this experiment is not

A/R: Thank you for this valuable comment and suggestion. The authors have tried to explain the queries in the revised manuscript as suggested by the reviewer. Ag was used only as a doping agent.

Reviewer 3 Report

The aim of this paper is to introduce a new type of sorbent based on silver nanoparticles and montmorillonite clay for the adsorption of methylene blue.

In my opinion the manuscript, while scientifically (mostly) sound, also has no great scientific value. From the data given the values determined for adsorption are not much better than MMT clay by itself, while the silver nitrate used adds a significant cost component. The English shows many mistakes but gets better in the second part of the article. There are huge problems with references. While the introduction shows a good amount of references, the references themselves are mostly wrong, or wrongly placed, which needs to be improved. Silver-NP/montmorillonite nanocomposites have been synthesized before, although not by this method. The very basic characterizations for adsorption on this material have been done, but things like kinetics, isotherms and (for me) most importantly desorption/reuse studies are missing. On the positive side, the methods section is quite well described.

Some more detailed comments follow below:

English: The English used in this paper contains a large number of small mistakes, almost in every sentence, either from a grammar or style point of view (mainly in the first part of the manuscript). Fortunately the manuscript is still readable despite this, but it should still be improved. There are too many mistakes to list them here.

Introduction:

-sentence “Moreover, synthetic dyes have a complex molecular structure which provides the stability  to common redox processes and makes them persistent in the environment [4,5].”: I do not think this sentence nor the references are quite correct here. While one reference basically just makes a statement and cites another reference (in which case you should have used that reference instead), the other in my opinion states the opposite if anything. I know from my own experience that especially methylene blue can degrade on its own, and other dyes can be degraded biologically for example (as an example reference see the “anaerobic treatment” part in the reference V.K.Gupta, Suhas, Application of low-cost adsorbents for dye removal – A review, Journal of Environmental Management, 90(8), 2009, 2313-2342). I would like you to replace this sentence with something else, name specific examples of very stable dyes (with appropriate references) or delete the sentence

- the next sentence “Consequently, some metal complex-based dyes are severely causing mutagenic and other health issues even at ppm concentrations” is not referenced correctly. The reference is about methylene blue and not complex based dyes. Please change the reference to some medical study that actually proves your point.

-“Methylene Blue (MB) is one of the widely used cationic dyes in textile industries like textile, paper, plastic and medicine because of its faster adsorption on cotton fabric, and economic favour [7]” again, the reference does not prove this; please list a reference that actually proves the point of the sentence

-“Whereas, the MB has several health issues such as cyanosis, jaundice, anaemia, hypertension, and tissue necrosis due to the MB dye active radical [8].” again, the reference does not prove this at all. It says nothing about radicals, it does list health problems but does not prove it but instead references something else. Normally, you should use that reference instead of the one you used except in this case the reference is about something completely different (malachite green). hence, please change the reference to something that actually proves the point you are making

-C16H18N3SCl: please put the numbers in subscript, like for a normal chemical formula; please also check the whole article for other formulas with this problem, I saw this multiple times with different formulas). Also please do not split the formula over two lines

-“There are several physicals, chemical, and biological methods for the removal of MB dye molecules from the water systems [9, 10]” again, the references are badly chosen as they are not about MB removal. Please use different references here

-“Therefore, adsorption  techniques are being used by  the  industries because of their low cost and easy handling [15–17].” these references are about research level studies. You can keep them if you want, but please add a reference that actually proves that adsorption techniques are used in industry for waste water purification

-“Among  the various available adsorbents, clays are abundantly available, natural and  low-cost materials having  excellent cation exchange efficiency due  to high surface area, mechanical stability, swelling properties and high cation exchange capacity (CEC).” a reference would be nice here

-please write a paragraph about the plant Sida Acuta, especially why it was chosen in the present study (cheap? waste material? specific properties in preparing silver nanoparticles that other plants do not have?), also with references

-2.2: you cannot call this “Biosynthesis”, choose something else like green synthesis, environmentally friendly synthesis or whatever else. However biosynthesis means something else and as such should not be used here

-could you please give a more detailed name for the UV and IR devices? Googling them (“UV analytical 002” and “SP 65, Perkin Elmer”) gave me no results. If they are too old to be easy to find via google please tell me this and give me a more detailed name anyway. I could not find the Malvern Nano S50 either

-FTIR: Wanyika et al, 2016 should get a reference (I think it is ref [30]?); also, the band at 1675 is not an OH stretch band, those would all be in the 3000+ region

-EDX: why isn’t potassium included in the table? I can see it in the EDX spectrum

-Surface area measurement: I think you should put the comparison values for surface area and pore size of the MMT clay in the article. I do not think you necessarily need to measure them yourself, they could probably be found in the literature already, as MMT is a commonly used material

-Figure 7: the lines going through the measured points, especially of the first 2 curves of fig. 7a, look a bit weird. I am pretty sure that the adsorption would not go down after going up first and that this is just due to a measuring error. I think you should remove the lines connecting the measurement points from the graph or replace them with a curve calculated by the least squares method. This would have been avoidable if you actually included a kinetics calculation. Also without actual numbers the last sentence that the MSN was more effective than MMT is useless, especially in the light that the graphs (and the following figure 8) suggest that they are about the same. I think you should cut out this sentence

-effect of initial MB concentration: personally, I think a percentage value for adsorption on the y axis would be more interesting than an absolute value. There you would also be seeing a downward trend instead of an upward one. I also think some more data points (with higher concentrations than 200 mg/l) would have been good, to show the adsorption capacity a bit more clearly. The last reference in the paragraph is not listed with a number (it is in fact not in the reference list at all) and needs to be added

-conclusion: the unit should be m²/g (you did not put superscript and abbreviated gram as gm)

-For me it is ok that you did not determine kinetics or isotherms, but I think desorption/reuse studies should definitely be added to this manuscript. If you synthesize a material that is already quite expensive due to addition of silver, you should at least prove that you can reuse it

Author Response

The aim of this paper is to introduce a new type of sorbent based on silver nanoparticles and montmorillonite clay for the adsorption of methylene blue.

In my opinion the manuscript, while scientifically (mostly) sound, also has no great scientific value. From the data given the values determined for adsorption are not much better than MMT clay by itself, while the silver nitrate used adds a significant cost component. The English shows many mistakes but gets better in the second part of the article. There are huge problems with references. While the introduction shows a good amount of references, the references themselves are mostly wrong, or wrongly placed, which needs to be improved. Silver-NP/montmorillonite nanocomposites have been synthesized before, although not by this method. The very basic characterizations for adsorption on this material have been done, but things like kinetics, isotherms and (for me) most importantly desorption/reuse studies are missing. On the positive side, the methods section is quite well described.

Some more detailed comments follow below:

English: The English used in this paper contains a large number of small mistakes, almost in every sentence, either from a grammar or style point of view (mainly in the first part of the manuscript). Fortunately the manuscript is still readable despite this, but it should still be improved. There are too many mistakes to list them here.

Introduction:

-sentence “Moreover, synthetic dyes have a complex molecular structure which provides the stability to common redox processes and makes them persistent in the environment [4,5].”: I do not think this sentence nor the references are quite correct here. While one reference basically just makes a statement and cites another reference (in which case you should have used that reference instead), the other in my opinion states the opposite if anything. I know from my own experience that especially methylene blue can degrade on its own, and other dyes can be degraded biologically for example (as an example reference see the “anaerobic treatment” part in the reference V.K.Gupta, Suhas, Application of low-cost adsorbents for dye removal – A review, Journal of Environmental Management, 90(8), 2009, 2313-2342). I would like you to replace this sentence with something else, name specific examples of very stable dyes (with appropriate references) or delete the sentence.

  • The sentence has been replaced (highlighted in yellow).

- the next sentence “Consequently, some metal complex-based dyes are severely causing mutagenic and other health issues even at ppm concentrations” is not referenced correctly. The reference is about methylene blue and not complex based dyes. Please change the reference to some medical study that actually proves your point.

  • The correct reference has been added.

-“Methylene Blue (MB) is one of the widely used cationic dyes in textile industries like textile, paper, plastic and medicine because of its faster adsorption on cotton fabric, and economic favour [7]” again, the reference does not prove this; please list a reference that actually proves the point of the sentence

  • The correct reference has been added.

-“Whereas, the MB has several health issues such as cyanosis, jaundice, anaemia, hypertension, and tissue necrosis due to the MB dye active radical [8].” again, the reference does not prove this at all. It says nothing about radicals, it does list health problems but does not prove it but instead references something else. Normally, you should use that reference instead of the one you used except in this case the reference is about something completely different (malachite green). hence, please change the reference to something that actually proves the point you are making

  • The correct reference has been added.

-C16H18N3SCl: please put the numbers in subscript, like for a normal chemical formula; please also check the whole article for other formulas with this problem, I saw this multiple times with different formulas). Also please do not split the formula over two lines

  • Changes made as per suggestion.

-“There are several physicals, chemical, and biological methods for the removal of MB dye molecules from the water systems [9, 10]” again, the references are badly chosen as they are not about MB removal. Please use different references here

  • The correct reference has been added.

-“Therefore, adsorption  techniques are being used by  the  industries because of their low cost and easy handling [15–17].” these references are about research level studies. You can keep them if you want, but please add a reference that actually proves that adsorption techniques are used in industry for waste water purification

  • The correct reference has been added.

-“Among  the various available adsorbents, clays are abundantly available, natural and  low-cost materials having  excellent cation exchange efficiency due  to high surface area, mechanical stability, swelling properties and high cation exchange capacity (CEC).” a reference would be nice here

  • Reference provided.

-please write a paragraph about the plant Sida Acuta, especially why it was chosen in the present study (cheap? waste material? specific properties in preparing silver nanoparticles that other plants do not have?), also with references

  • A paragraph about Sida acuta weed plant used in this study has been added.

-2.2: you cannot call this “Biosynthesis”, choose something else like green synthesis, environmentally friendly synthesis or whatever else. However biosynthesis means something else and as such should not be used here

  • The term ‘Biosynthesis’ has been replaced with ‘Synthesis’ as per suggestion.

-could you please give a more detailed name for the UV and IR devices? Googling them (“UV analytical 002” and “SP 65, Perkin Elmer”) gave me no results. If they are too old to be easy to find via google please tell me this and give me a more detailed name anyway. I could not find the Malvern Nano S50 either

-FTIR: Wanyika et al, 2016 should get a reference (I think it is ref [30]?); also, the band at 1675 is not an OH stretch band, those would all be in the 3000+ region

  • Changes made as per suggestion.

-EDX: why isn’t potassium included in the table? I can see it in the EDX spectrum

-Surface area measurement: I think you should put the comparison values for surface area and pore size of the MMT clay in the article. I do not think you necessarily need to measure them yourself, they could probably be found in the literature already, as MMT is a commonly used material

  • BET data removed.

-Figure 7: the lines going through the measured points, especially of the first 2 curves of fig. 7a, look a bit weird. I am pretty sure that the adsorption would not go down after going up first and that this is just due to a measuring error. I think you should remove the lines connecting the measurement points from the graph or replace them with a curve calculated by the least squares method. This would have been avoidable if you actually included a kinetics calculation. Also without actual numbers the last sentence that the MSN was more effective than MMT is useless, especially in the light that the graphs (and the following figure 8) suggest that they are about the same. I think you should cut out this sentence

  • We have re-written the discussion section involving MB removal study.

-effect of initial MB concentration: personally, I think a percentage value for adsorption on the y axis would be more interesting than an absolute value. There you would also be seeing a downward trend instead of an upward one. I also think some more data points (with higher concentrations than 200 mg/l) would have been good, to show the adsorption capacity a bit more clearly. The last reference in the paragraph is not listed with a number (it is in fact not in the reference list at all) and needs to be added

  • Changes has been made as per suggestion.

-conclusion: the unit should be m²/g (you did not put superscript and abbreviated gram as gm)

  • The unit m2/gm has been replaced with m²/g.

-For me it is ok that you did not determine kinetics or isotherms, but I think desorption/reuse studies should definitely be added to this manuscript. If you synthesize a material that is already quite expensive due to addition of silver, you should at least prove that you can reuse it

  • Changes made as per suggestion.

Round 2

Reviewer 2 Report

Many thanks for throughly revising the manuscript which appears much more consistent and comprehensive. Therefore, the manuscript is recommended for acceptance.  However, some text editing (e.g. repetitions, spelling etc.) would be appreciated.

Author Response

  1. Many thanks for thoroughly revising the manuscript which appears much more consistent and comprehensive. Therefore, the manuscript is recommended for acceptance.  However, some text editing (e.g. repetitions, spelling etc.) would be appreciated.

A/R: Thank you for accepting the manuscript. The authors have now thoroughly checked the repetitions, spelling and grammatical errors in the revised manuscript. The English corrections are marked in yellow highlights in the whole manuscript

Reviewer 3 Report

-I guess you mainly rectified the wrong references by deleting the corresponding text passages. While I am not really happy about it, I suppose it works.

-reference 4 still only references other references. You should cite original documents whenever possible. In this case, after checking the reference in ref 4 I came up with this link “The United States Pharmacopeial Convention. Methylene Blue. Retrieved from https://cdn.ymaws.com/www.aavpt.org/resource/resmgr/imported/methyleneBlue.pdf “ which you can cite instead of ref 4. Before you do so please make sure that it works for you though, because the link that was cited did not exactly work for me

-“Therefore, adsorption techniques are being used by the industries because of low cost and easy handling [12-14].” still not the type of references I am actually looking for, but I suppose reviews are better than research level articles for this

-could you please give a more detailed name for the UV and IR devices? Googling them (“UV analytical 002” and “SP 65, Perkin Elmer”) gave me no results. If they are too old to be easy to find via google please tell me this and give me a more detailed name anyway. I could not find the Malvern Nano S50 either<—it seems you ignored this. Since there are no more detailed names, could you maybe provide me with internet links or something so that I can see what devices these are?

-in the new Fig. 8, was the data point that I was commenting on in my last review remeasured? or how did the change in values happen?

The authors have addressed most of the points I wrote in my review (although the desorption/reuse study was not added, which I would consider important). However, I do not like the way that some of the article was changed at all, mostly that all the data on pure MMT clay was taken out. At present I am leaning more towards rejecting the article, however if the editor decides to keep the article I would suggest for the authors to put the comparative data with MMT back in (and the BET measurements with a comparison to MMT), address the problem in the conclusion (i.e. that the adsorption efficiency is not much better for MSN than for regular MMT) and do a complete check of the English used in the article by a fluent English speaker.

Author Response

  1. I guess you mainly rectified the wrong references by deleting the corresponding text passages. While I am not really happy about it, I suppose it works.

A/R: Thank you for your comment and suggestion. We have made changes in this revised manuscript as per reviewer’s suggestions by rewriting some paragraphs in introduction section, by providing relevant references and by rediscussing the MB dye removal section (where we are focusing on MB removal only by nanocomposite material).

  1. -reference 4 still only references other references. You should cite original documents whenever possible. In this case, after checking the reference in ref 4 I came up with this link “The United States Pharmacopeial Convention. Methylene Blue. Retrieved from https://cdn.ymaws.com/www.aavpt.org/resource/resmgr/imported/methyleneBlue.pdf “ which you can cite instead of ref 4. Before you do so please make sure that it works for you though, because the link that was cited did not exactly work for me

A/R: Thank you for your comment and suggestion, Sir. But I am unable to understand, please elaborate. The reference 4 (Tsade Kara, H., Anshebo, S.T., Sabir, F.K. and Adam Workineh, G., 2021. Removal of Methylene Blue Dye from Wastewater Using Periodiated Modified Nanocellulose. International Journal of Chemical Engineering, 2021.) mentions toxic sides of MB dye. Further reference 5 supports the adverse effects of MB on CNS.

  1. -“Therefore, adsorption techniques are being used by the industries because of low cost and easy handling [12-14].” still not the type of references I am actually looking for, but I suppose reviews are better than research level articles for this

A/R: Thank you for your comment and suggestion. The authors have now provided relevant references in the revised manuscript [Ref. 12-14]. Hope this would satisfy the query raised by the reviewer.

  1. -could you please give a more detailed name for the UV and IR devices? Googling them (“UV analytical 002” and “SP 65, Perkin Elmer”) gave me no results. If they are too old to be easy to find via google please tell me this and give me a more detailed name anyway. I could not find the Malvern Nano S50 either<—it seems you ignored this. Since there are no more detailed names, could you maybe provide me with internet links or something so that I can see what devices these are?

A/R: Thank you for this valuable comment and suggestion. The authors have now provided the detailed and correct description of the required instruments in the revised manuscript as suggested by the reviewer.

Actually it was mistakes from our side, which is rectified now as following

Earlier model name

Correct model name

1

UV- analytical 002

2060+ Analytical

2

FTIR- SP 65, Perkin Elmer

Perkin Elmer, ‘Spectrum 6500’ (USA)

3

Malvern Nano S50

Malvern Zetasizer, Z-90 (UK)

We have provided link for the instruments we have used from our central instrumentation facility (CIF), CUG, Gandhinagar.

  1. -in the new Fig. 8, was the data point that I was commenting on in my last review remeasured? or how did the change in values happen?

A/R: Thank you for this valuable comment and suggestion. Fig. 8 is not a new one. It is the same figure 7(b) in the initial manuscript and you did not make any comment on fig 7(b) in your last review.

  1. The authors have addressed most of the points I wrote in my review (although the desorption/reuse study was not added, which I would consider important). However, I do not like the way that some of the article was changed at all, mostly that all the data on pure MMT clay was taken out. At present I am leaning more towards rejecting the article, however if the editor decides to keep the article I would suggest for the authors to put the comparative data with MMT back in (and the BET measurements with a comparison to MMT), address the problem in the conclusion (i.e. that the adsorption efficiency is not much better for MSN than for regular MMT) and do a complete check of the English used in the article by a fluent English speaker.

 A/R: The authors have now thoroughly revised the manuscript in terms of English editing. All the characterization data (including BET) associated with raw MMT has been added. The reviewer must take a look over it and hope will be satisfied by the revised version of the manuscript.

Round 3

Reviewer 3 Report

A/R: Thank you for your comment and suggestion, Sir. But I am unable to understand, please elaborate.

-There are multiple problems that play a role here in my opinion: The first problem is that not every reader reads an article for the main information that the article is meant to convey, but for other information that it also imparts. An example from organic chemistry: Let’s say there is an article about a new method for stereoselective epoxidation. As part of the article, several substrates have been tested for this epoxidation; those substrates also have to be synthesized first (or bought from a commercial supplier), which the authors would obviously have to have done beforehand. Now there will be some readers of the article who do not actually care about the epoxidation, but only want to find a way to synthesize the substrate. If the authors publish the full synthesis procedure, all is good. If they only write that it has been done before, and give a reference, the reader can look up the reference and find the procedure there. But if that reference lists another reference, and the reference lists another, etc etc, then the reader is left digging through a lot of articles before they can get the actual information they are looking for. In the worst case the final reference would state something like the substrate was commercially available(and the product has now been discontinued) or that it was a gift from someone, or it just does not list the synthesis procedure. If the authors of the first article had been more diligent and actually looked up the reference they could have noticed that and done this work themselves. In fact they should have done this work themselves, and then if they found that there is not actually a synthesis procedure listed, they could have written the procedure they listed.

Another possibility would be that the meaning gets somewhat distorted through following the references. Authors are not infallible, they are also only humans and can make mistakes. I have recently seen in a manuscript a claim about crosslinking of alginate with different metals, and whether the bond formed is more covalent or more ionic. In the manuscript they wanted to compare Fe2+ and Fe3+, but as I went through the reference they listed and then another reference that was listed in that reference it came out that the original information said nothing about Fe2+, but Ca2+, which is completely different. However, the authors only read what they wanted to and changed the information in their minds. If they had cited the original article in their manuscript, it would have been easy to see.

Yet another possibility would be that the original information is plain wrong, but has only been corrected later and whoever used the information as reference in their article did not see the correction yet (and whoever cites the new article will never see that the first article has been retracted if they do not follow the chain of references). This means that a wrong fact can get perpetuated through citations and more citations if noone bothers to check it.

https://www.themandarin.com.au/157665-academic-journals-journalists-perpetuate-misinformation-in-their-handling-of-research-retractions-a-new-study-finds/

Here is a web article that, while not exactly on the subject, shows that wrong statements can get also perpetuated by citations and chains of references. If you cite an article that cites an article that cites an article that cites a retracted article, unless you follow this chain of citations, you (as authors of a new manuscript) would not notice that the information you cite is actually wrong.

Finally, some authors just make statements and do not cite anything at all, maybe because they couldn’t find a good reference, maybe because they believe “everybody knows that”, or for other reasons. When someone else cites their paper for this information, and you (as authors of a new manuscript) cite the new article, unless you actually follow the chain of references nothing looks amiss.

These reasons are why I would like to ask you (and every other author as well) to cite the original source when making a statement, or at least a very reputable one (like official government documents as in this case or like a review (because I think/hope people who write a review actually check references a bit better)).

A/R: Thank you for this valuable comment and suggestion. Fig. 8 is not a new one. It is the same figure 7(b) in the initial manuscript and you did not make any comment on fig 7(b) in your last review.

-I am sorry, you are right, my comment was on Fig. 7a, which you omitted in the last version

A/R: The authors have now thoroughly revised the manuscript in terms of English editing. All the characterization data (including BET) associated with raw MMT has been added. The reviewer must take a look over it and hope will be satisfied by the revised version of the manuscript.

-the English has definitely been improved, although there are still a few mistakes left in the manuscript. I would like to thank the authors for including some of the characterization (BET data) again, however, all the adsorption characteristics for MMT that used to be there as a comparison is still missing. I do consider them important information, as it basically shows that (at least for use as an adsorbent) the material described in this manuscript is not really useful, from a cost/benefit point of view. There are also no studies done regarding desorption/reuse, as suggested by my first review. However, I can also see that the authors are in a difficult situation regarding the reviewers, as the first reviews ranged from “basically ok”, “you need to do more” to “reject”, then when you do cut out some data the reviewer who gave you a reject liked it better while the other one (me) did not agree with cutting it out, so you basically cannot change it in a way that makes all reviewers happy. At this point I will let the scientific editor decide it: I am still thinking that this article should include the comparison adsorption data between MSN and MMT, as well as desorption/reuse studies or an explanation why this is not necessary (in my opinion the material is too expensive for using it only once, so I think there needs to be a reuse planned), however if the scientific editor decides the manuscript is ok even without this then I will not insist on it anymore.

Author Response

A/R: Thank you for your comment and suggestion, Sir. But I am unable to understand, please elaborate.

-There are multiple problems that play a role here in my opinion: The first problem is that not every reader reads an article for the main information that the article is meant to convey, but for other information that it also imparts. An example from organic chemistry: Let’s say there is an article about a new method for stereoselective epoxidation. As part of the article, several substrates have been tested for this epoxidation; those substrates also have to be synthesized first (or bought from a commercial supplier), which the authors would obviously have to have done beforehand. Now there will be some readers of the article who do not actually care about the epoxidation, but only want to find a way to synthesize the substrate. If the authors publish the full synthesis procedure, all is good. If they only write that it has been done before, and give a reference, the reader can look up the reference and find the procedure there. But if that reference lists another reference, and the reference lists another, etc etc, then the reader is left digging through a lot of articles before they can get the actual information they are looking for. In the worst case the final reference would state something like the substrate was commercially available(and the product has now been discontinued) or that it was a gift from someone, or it just does not list the synthesis procedure. If the authors of the first article had been more diligent and actually looked up the reference they could have noticed that and done this work themselves. In fact they should have done this work themselves, and then if they found that there is not actually a synthesis procedure listed, they could have written the procedure they listed.

Another possibility would be that the meaning gets somewhat distorted through following the references. Authors are not infallible, they are also only humans and can make mistakes. I have recently seen in a manuscript a claim about crosslinking of alginate with different metals, and whether the bond formed is more covalent or more ionic. In the manuscript they wanted to compare Fe2+ and Fe3+, but as I went through the reference they listed and then another reference that was listed in that reference it came out that the original information said nothing about Fe2+, but Ca2+, which is completely different. However, the authors only read what they wanted to and changed the information in their minds. If they had cited the original article in their manuscript, it would have been easy to see.

Yet another possibility would be that the original information is plain wrong, but has only been corrected later and whoever used the information as reference in their article did not see the correction yet (and whoever cites the new article will never see that the first article has been retracted if they do not follow the chain of references). This means that a wrong fact can get perpetuated through citations and more citations if noone bothers to check it.

https://www.themandarin.com.au/157665-academic-journals-journalists-perpetuate-misinformation-in-their-handling-of-research-retractions-a-new-study-finds/

Here is a web article that, while not exactly on the subject, shows that wrong statements can get also perpetuated by citations and chains of references. If you cite an article that cites an article that cites an article that cites a retracted article, unless you follow this chain of citations, you (as authors of a new manuscript) would not notice that the information you cite is actually wrong.

Finally, some authors just make statements and do not cite anything at all, maybe because they couldn’t find a good reference, maybe because they believe “everybody knows that”, or for other reasons. When someone else cites their paper for this information, and you (as authors of a new manuscript) cite the new article, unless you actually follow the chain of references nothing looks amiss.

These reasons are why I would like to ask you (and every other author as well) to cite the original source when making a statement, or at least a very reputable one (like official government documents as in this case or like a review (because I think/hope people who write a review actually check references a bit better)).

Thank you, Sir, for your kind explanation and suggestion. I understood your point. Change in reference has been made accordingly.

A/R: Thank you for this valuable comment and suggestion. Fig. 8 is not a new one. It is the same figure 7(b) in the initial manuscript and you did not make any comment on fig 7(b) in your last review.

-I am sorry, you are right, my comment was on Fig. 7a, which you omitted in the last version

A/R: The authors have now thoroughly revised the manuscript in terms of English editing. All the characterization data (including BET) associated with raw MMT has been added. The reviewer must take a look over it and hope will be satisfied by the revised version of the manuscript.

-the English has definitely been improved, although there are still a few mistakes left in the manuscript. I would like to thank the authors for including some of the characterization (BET data) again, however, all the adsorption characteristics for MMT that used to be there as a comparison is still missing. I do consider them important information, as it basically shows that (at least for use as an adsorbent) the material described in this manuscript is not really useful, from a cost/benefit point of view. There are also no studies done regarding desorption/reuse, as suggested by my first review. However, I can also see that the authors are in a difficult situation regarding the reviewers, as the first reviews ranged from “basically ok”, “you need to do more” to “reject”, then when you do cut out some data the reviewer who gave you a reject liked it better while the other one (me) did not agree with cutting it out, so you basically cannot change it in a way that makes all reviewers happy. At this point I will let the scientific editor decide it: I am still thinking that this article should include the comparison adsorption data between MSN and MMT, as well as desorption/reuse studies or an explanation why this is not necessary (in my opinion the material is too expensive for using it only once, so I think there needs to be reuse planned), however, if the scientific editor decides the manuscript is ok even without this then I will not insist on it anymore.

Thank you for your comment and suggestion. We have added comparative adsorption data as per your suggestion.